# Cellular aspect ratio and cell division mechanics underlie the patterning of cell progeny in diverse mammalian epithelia

Kara L McKinley[1,2], Nico Stuurman[1,2], Loic A Royer[3], Christoph Schartner[4], David Castillo-Azofeifa[5,6], Markus Delling[4], Ophir D Klein[5,6,7,8]*, Ronald D Vale[1,2]*

[1]Department of Cellular and Molecular Pharmacology, University of California, San Francisco, San Francisco, United States; [2]Howard Hughes Medical Institute, University of California, San Francisco, San Francisco, United States; [3]Chan Zuckerberg Biohub, San Francisco, United States; [4]Department of Physiology, University of California, San Francisco, San Francisco, United States; [5]Department of Orofacial Sciences, University of California, San Francisco, San Francisco, United States; [6]Program in Craniofacial Biology, University of California, San Francisco, San Francisco, United States; [7]Department of Pediatrics, University of California, San Francisco, San Francisco, United States; [8]Institute for Human Genetics, University of California, San Francisco, San Francisco, United States

**\*For correspondence:**
Ophir.Klein@ucsf.edu (ODK);
Ron.Vale@ucsf.edu (RDV)

**Competing interests:** The authors declare that no competing interests exist.

**Abstract** Cell division is essential to expand, shape, and replenish epithelia. In the adult small intestine, cells from a common progenitor intermix with other lineages, whereas cell progeny in many other epithelia form contiguous patches. The mechanisms that generate these distinct patterns of progeny are poorly understood. Using light sheet and confocal imaging of intestinal organoids, we show that lineages intersperse during cytokinesis, when elongated interphase cells insert between apically displaced daughters. Reducing the cellular aspect ratio to minimize the height difference between interphase and mitotic cells disrupts interspersion, producing contiguous patches. Cellular aspect ratio is similarly a key parameter for division-coupled interspersion in the early mouse embryo, suggesting that this physical mechanism for patterning progeny may pertain to many mammalian epithelia. Our results reveal that the process of cytokinesis in elongated mammalian epithelia allows lineages to intermix and that cellular aspect ratio is a critical modulator of the progeny pattern.
DOI: https://doi.org/10.7554/eLife.36739.001

## Introduction

Epithelia are sheets of polarized cells that function as barriers between compartments of multicellular organisms and between the organism and the external environment. In addition to providing a physical barrier, specialized epithelial cell types provide functions including sensation, absorption and secretion, and contribute to the identities of nearby cells through cell-cell signaling. Proper epithelial function requires that these diverse cell types are positioned appropriately within the tissue and that this distribution is maintained as new cells are added through cell division.

The adult mammalian small intestine is a prime example of an epithelium that contains many cell types and maintains a high degree of spatial organization during rapid turnover (*Barker, 2014*). In the small intestine, divisions of stem cells in the crypts of Lieberkühn replenish the stem cell pool and generate absorptive and secretory progenitor cells in the crypt, which in turn produce differentiated cells that carry out the absorptive and protective functions of the gut (*Gracz and Magness, 2014*). Throughout the epithelium, cells derived from a given progenitor intersperse with other cells

**eLife digest** The body has an impressive ability to renew itself by replacing old and damaged cells with new ones. This can happen rapidly; for example, the lining of the intestine renews itself approximately every five days. The lining contains many different cell types, which exchange important signals with their neighbors. This means that the new cells need to occupy similar positions to the ones they are replacing to keep the intestine working.

New cells form when existing cells double their contents and divide. In many tissues the resulting cells sit side-by-side. But when cells in the intestine divide, the new cells often separate, ending up on either side of a cell that did not divide.

To investigate how this happens, McKinley et al. used live microscopy techniques to watch in real time as new cells divide and position themselves in mouse intestinal organoids – miniature versions of organs that can be grown outside the body. This revealed that the shape of intestinal cells explains why the newly formed cells become separated. Intestinal cells are taller than they are wide, and divide near their top edge. This enables a neighboring cell to squeeze between the new cells as they divide.

Further experiments showed that tall cells in other mouse tissues also become separated after division. The process of new cells interspersing with their neighbors due to their height is therefore not unique to the intestine. It may also be common in other mammalian tissues. There is great potential for investigating this further because labs can now grow many types of organoids, representing different organs. Using live microscopy to examine them could reveal more about how various tissues grow.

DOI: https://doi.org/10.7554/eLife.36739.002

(*Carroll et al., 2017*). In particular, lineage tracing in fixed tissues has established that cells derived from secretory progenitors intermix with cells derived from absorptive progenitors along the crypt and villus length (*Yang et al., 2001*). At the crypt base, stem cells are interspersed with Paneth cells (*Farin et al., 2016*). Interspersion of cell lineages plays important roles in determining local signaling environments required for intestinal homeostasis. For example, intestinal stem cells receive signals critical to their identity from neighboring Paneth cells (*Sato et al., 2011*). Indeed, direct contact between stem and Paneth cells supports stem cell maintenance (*Farin et al., 2016*). However, the molecular mechanisms that underlie the intermixing of lineages are poorly understood.

Here, we use light sheet and confocal imaging of live murine small intestinal organoids to define the mechanisms of cell interspersion. We find that rearrangements of the actin cytoskeleton displace mitotic cells along the apical-basal axis, such that cell division occurs at the apical surface. Interspersion arises when elongated interphase neighboring cells wedge between apically dividing daughters during cytokinesis. We find that the propensity to intersperse during division requires an elongated shape of cells in the epithelium; reducing the cellular aspect ratio (height: width) in organoids disrupts interspersion, resulting in outgrowth of lineage patches. Consistent with our data indicating that the physical parameters of the tissue are a critical determinant of interspersion during division, we demonstrate that the elongated epiblast/primitive ectoderm of post-implantation (E7.5) mouse embryos, but not the short visceral endoderm, also undergoes division-coupled cell interspersion. Thus, tissues of distinct developmental context from the adult small intestine exhibit similar mechanisms for patterning cellular progeny according to cellular dimensions. Together, our data indicate that cell shape differences between interphase and mitotic cells in elongated mammalian epithelia can allow a neighboring cell to insert between nascent daughter cells during cytokinesis and drive interspersion of cellular progeny.

## Results

### Cells of different lineages intersperse during cell division in the intestinal epithelium

To identify the basis for cell interspersion, we performed time-lapse imaging of adult murine small intestinal organoids (*Kretzschmar and Clevers, 2016*; *Sato et al., 2009*) by confocal and light sheet

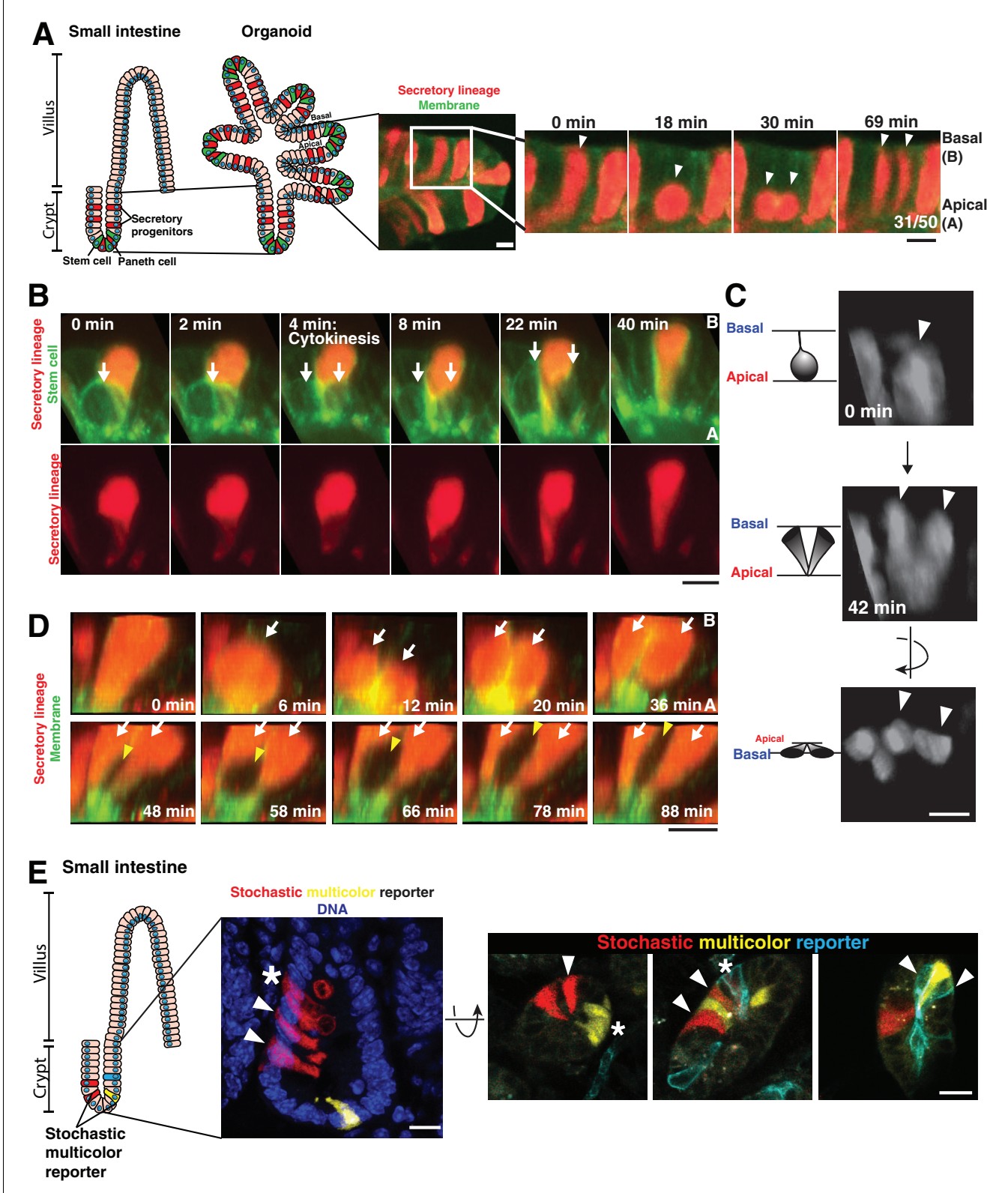

**Figure 1.** Separation of dividing daughter cells during apical cytokinesis underlies intermingling of cell lineages. (**A**) Left: Cartoon depicting organoid derivation. Right: Frames from time-lapse imaging of a dividing cell of the secretory lineage (red, $Atoh1^{CreER}$; $R26^{RFP}$) interspersing with non-secretory cells (green membranes). Arrowheads: dividing cell. Fraction of divisions in which labeled daughters separated is shown on the right panel. (**B**) Frames from 3D reconstructed SPIM of a secretory cell (red, $Atoh1^{CreER}$; $R26^{RFP}$) inserting in the cytokinetic furrow of a dividing stem cell (green, $Lgr5^{DTR-GFP}$). *Figure 1 continued on next page*

*Figure 1 continued*

Arrows: dividing cell. (C) Frames from 3D reconstructed SPIM of a dividing cell of the secretory lineage (*Atoh1^CreER^; R26^RFP^*). Arrowheads: dividing cell. (D) Frames from 3D reconstructed SPIM of a secretory cell (red) undergoing a division in which daughter cells do not separate during cytokinesis (top, white arrows indicate daughter cells). Subsequently, these daughter cells become separated by a dividing cell pushing between them (bottom, white arrows indicate daughter cells and yellow arrowhead indicates newly dividing cell inserting between the adjacent daughters). (E) Confocal images of crypts in which cells have been labeled with a stochastic multicolor reporter in vivo (*Vil1^CreER^; R26^Brainbow2.1^*) and the positions of progeny analyzed three days after induction of the reporter. Left: sagittal view from 50 µm sections. Right: transverse views from 20 µm sections. Arrowheads indicate interspersed progeny. Progeny can also remain adjacent, as in the organoids, indicated by asterisks. Scale bars, 10 µm.

DOI: https://doi.org/10.7554/eLife.36739.003

The following video and figure supplement are available for figure 1:

**Figure supplement 1.** Cell interspersion in intestinal organoids.

DOI: https://doi.org/10.7554/eLife.36739.004

**Figure 1—video 1.** 3D reconstruction of separated daughter cells.

DOI: https://doi.org/10.7554/eLife.36739.005

**Figure 1—video 2.** Daughters that remain neighbors can become separated by subsequent nearby mitosis.

DOI: https://doi.org/10.7554/eLife.36739.006

microscopy (single plane illumination microscopy - SPIM) (*Wu et al., 2013*) (*Figure 1A*). To visualize cell lineages, we first used organoids in which the cytoplasm of cells of the secretory lineage was labeled with RFP (*Atoh1^CreER^; R26^RFP^*). Strikingly, we observed that daughter cells separated from one another in approximately half of divisions (31/50 divisions, *Figure 1A* and *Video 1*; also see [*Carroll et al., 2017*]). We observed that Atoh1-expressing secretory daughters along the crypt length separated from one another, mixing with unlabeled cells (*Figure 1A* and *Video 1*). 3D SPIM data confirmed that cells were fully separated on their basal surface, although they maintained a minimal contact on the apical surface, creating a V-shaped geometry (*Figure 1C*, *Figure 1—figure supplement 1E*, *Figure 1—video 1*, 9/16 daughter pairs). When daughters did not separate during the division (*Figure 1D*, top panels, 7/16 daughter pairs), these daughters either became separated at later time points by division of a neighboring cell (*Figure 1D*, bottom panels and *Figure 1—video 2*), or remained as neighbors for the duration of imaging. These data indicate that separation of nascent daughter cells during cell division makes substantial contributions to the relative positioning of cell types within the intestinal epithelium.

We next tested whether daughter cell separation was a common feature of cell lineages in the intestinal epithelium. Notch1-expressing cells (from *Notch1^CreERT2^; R26^RFP^* organoids), which comprise all non-secretory cells including stem cells and absorptive cells, also interspersed during division (*Figure 1—figure supplement 1A*). Finally, dividing stem cells (labeled with *Lgr5^DTR-GFP^*) at the crypt base also separated, with secretory (Paneth) cells (labeled with *Atoh1^CreER^; R26^RFP^*) inserting between them (*Figure 1B*, *Figure 1—figure supplement 1B* and *Video 2*). Altering cell fates, for example by inhibiting Notch signaling to cause an expansion of secretory cells, did not alter the frequency of this process (*Figure 1—figure supplement 1C,D*). Thus, cells intersperse during a subset of divisions in all cell lineages of the crypt epithelium.

We next sought to determine whether the interspersion of cellular progeny observed in organoids also occurred in the intestine in vivo. To this end, we labeled a subset of cells in the intestines of adult mice with different fluorophores by induction of the stochastic multicolor reporter allele, *R26^Brainbow2.1^* (*Vil1^CreERT2^; R26^Brainbow2.1^*). After three days of Cre induction, which is sufficient for most crypt epithelial cells to divide at least once (*Snippert et al., 2010*), the intestines were fixed and the positions of progeny analyzed in thick sections. Consistent with our organoid imaging, we observed that a subset of progeny (18/40 progeny pairs, n = 3 mice) were interspersed with unlabeled cells or differently labeled cells in the intact intestine (*Figure 1E*). Thus, progeny

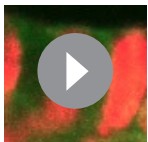

**Video 1.** Secretory cell separation during division. Cells of the secretory lineage (red, *Atoh1^CreER^; R26^RFP^*) interspersed with non-secretory cells (green membranes) imaged by spinning disc confocal with 20X objective at 3 min time points.

DOI: https://doi.org/10.7554/eLife.36739.007

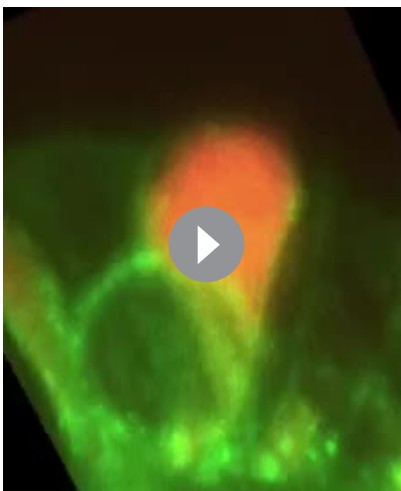

**Video 2.** Stem cell separation during division by insertion of a secretory cell into the cytokinetic furrow. Cell of the secretory lineage (red, *Atoh1*$^{CreER}$; *R26*$^{RFP}$) inserts into the furrow of a dividing stem cell (green, *Lgr5*$^{DTR-GFP}$). Imaged by SPIM with 40X objectives at 2 min time points. Second clip isolates only the cell of the secretory lineage.
DOI: https://doi.org/10.7554/eLife.36739.008

intersperse with neighboring cells in intestinal organoids and in the intestinal epithelium in vivo.

## Cells intersperse during cytokinesis as part of a suite of cell shape changes restricted to the basolateral surface by cell-cell contact

We next sought to characterize the cell behaviors that give rise to interspersion during cell division in the intestinal epithelium. We observed that mixing occurred as cells underwent cytokinesis on the apical surface of the epithelium, during which neighboring cells intruded within the ingressing cytokinetic furrow (*Figure 1B*, *Video 2*). First, mitotic cells displaced to the apical surface of the epithelium, and the dramatic reduction in their basal footprint caused neighboring cells to reposition and occupy the position above (basal to) the mitotic cell (*Figure 1B*, *Figure 1—figure supplement 1B*). Cells progressed through a polarized (non-concentric) cytokinesis (*Figure 2A*, *Video 2*, *Figure 2—videos 1*, *2* and *3*) (also see [*Fleming et al., 2007*]), in which the cleavage furrow initiated from the basal surface and then progressed to the apical surface. As cytokinesis continued, a minimal daughter-daughter contact remained on the apical surface (*Figure 1—figure supplement 1E*). We note that this minimal vertex contact is consistent with other reports of daughter cell geometry during vertebrate cytokinesis (*Higashi et al., 2016*), but contrasts with the long daughter-daughter interface generated during cytokinesis in *Drosophila* epithelia (*Gibson et al., 2006*; *Herszterg et al., 2013*; *Pinheiro et al., 2017*), as we will return to in the Discussion. The minimal contact between daughters generated by cytokinesis allowed a neighboring interphase cell to wedge between the daughters (*Video 2*). Finally, as the division completed, the daughter cells elongated on either side of the invading neighbor cell to occupy the full apical-basal axis in interphase (*Figure 1*, *Video 2*).

In contrast to the dramatic shape changes on the basal surface of dividing cells, the apical surface remained unperturbed: the apical footprint of the mitotic cell was similar to its interphase neighbors (*Figure 2—figure supplement 1A–C*), and a cytokinetic furrow was absent from the apical surface as in many metazoan epithelia (*Fleming et al., 2007*; *Guillot and Lecuit, 2013*; *Herszterg et al., 2013*; *Founounou et al., 2013*). Previous studies showed that cell-cell junctions on the apical surface of the intestine persist throughout mitosis (*Jinguji and Ishikawa, 1992*) and staining with junctional markers indicated that the same is true for intestinal organoids (*Figure 2—figure supplement 1A*). To test the possibility that persistent cell-cell contacts oppose mitotic shape changes on the apical surface, we dissociated organoids into single cells or pairs of cells and performed time-lapse imaging of mitotic exit. In contrast to the polarized cytokinesis that occurs in the tissue, cytokinesis occurred symmetrically in dissociated cells (*Figure 2B*, *Figure 2—video 4*), suggesting that tissue architecture plays a crucial role in this polarization. Together, these data indicate that mixing arises during cytokinesis as part of a suite of mitotic cell shape changes that are confined to the basolateral surface within the context of the tissue.

## Rearrangements of the actin cytoskeleton during cell division displace dividing cells along the apical-basal axis

Our observations suggested that a critical initiating step during cell interspersion was the positioning of the dividing cell on the apical surface of the epithelium. We therefore sought to determine the mechanism that gives rise to this apical displacement. Apical displacement initiated concurrently

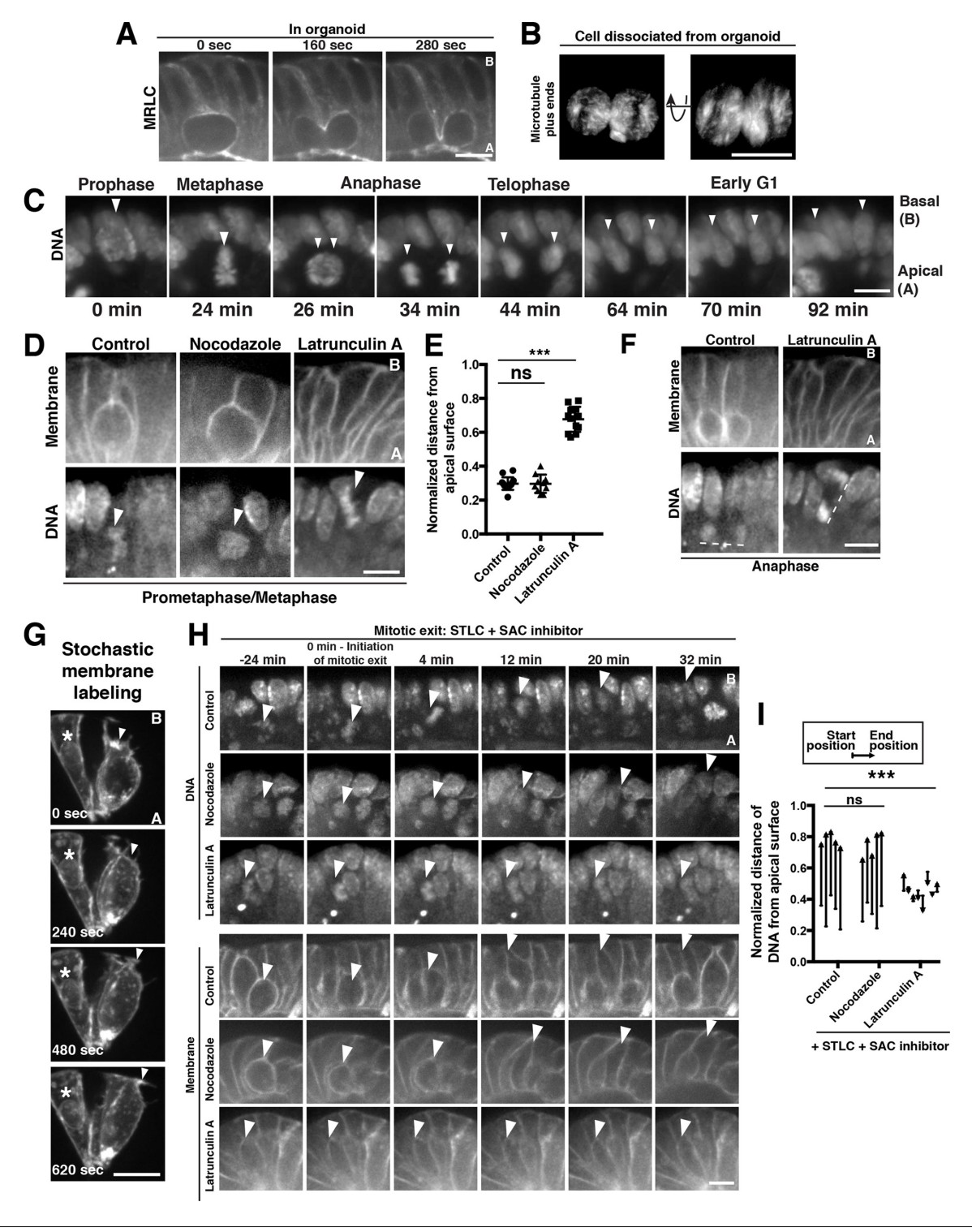

**Figure 2.** Polarized actin-dependent cell shape changes underlie division-coupled interspersion behaviors. (**A**) Frames from time-lapse imaging of cytokinesis in an organoid expressing myosin regulatory light chain (MRLC)-mScarlet. (**B**) 3D reconstruction from live imaging of a cell dissociated from EB3-GFP organoids undergoing cytokinesis. EB3-GFP labeled organoids were used to facilitate identification of dissociated cells undergoing mitosis. Representative of 12/15 divisions. (**C**) Frames from SPIM of chromosome segregation in a live organoid. DNA: H2B-mScarlet. Arrowheads indicate mitotic chromosome masses. (**D**) Frames from confocal imaging of mitotic cells in live organoids treated with cytoskeletal inhibitors for 30 min before initiation of imaging. Membranes: *R26^{mTmG}*; DNA: SiR-DNA. Arrowheads: mitotic chromosomes. (**E**) Quantification of the distance of mitotic chromosomes from the apical surface of the organoid epithelium following treatment with cytoskeletal inhibitors, normalized to the total apical-basal

*Figure 2 continued on next page*

*Figure 2 continued*

height of the epithelium, n ≥ 10. ns: not significant; ***p<0.001, Student's t-test. (F) Anaphase of mitoses shown in (D). Dashed lines underline anaphase chromosome masses. (G) Frames from time-lapse imaging of *Vil1^CreERT2*; *R26^mTmG* organoids in which recombination has been induced at low levels to label a subset of cell membranes in the organoid. The protrusive front of one daughter cell is indicated by an arrowhead. Note that the division occurred along the imaging plane, such that the other daughter cell is 'behind' the imaged daughter cell. Asterisk: nearby interphase cell that did not participate in the division. (H) Frames from confocal imaging of live organoids testing the cytoskeletal requirements for the basal movement of nascent nuclei (top, arrowheads indicate chromosomes) and elongation of the basal cell edge (bottom, arrowhead indicates basal edge of reinserting cell). A schematic of this experiment is shown in *Figure 2—figure supplement 1I*. DNA: SiR-DNA; Membrane: *R26^mTmG*; STLC: Eg5 inhibitor to induce mitotic arrest; SAC: spindle assembly checkpoint. (I) Quantification of DNA position before SAC inhibition (starting position), and at chromosome decondensation (end position), normalized to the total apical-basal distance of the epithelium. Arrowheads point towards the end position after mitotic exit. n ≥ 5, ns: not significant, ***: p<0.001, Student's t-test of comparing end position and start position. Scale bars, 10 μm.
DOI: https://doi.org/10.7554/eLife.36739.009

The following video and figure supplement are available for figure 2:

**Figure supplement 1.** Polarized actin-dependent changes in cell shape during division in intestinal organoids.
DOI: https://doi.org/10.7554/eLife.36739.010
**Figure 2—video 1.** Cytokinesis in the intestinal organoids.
DOI: https://doi.org/10.7554/eLife.36739.011
**Figure 2—video 2.** Cytokinesis in the intestinal organoids.
DOI: https://doi.org/10.7554/eLife.36739.012
**Figure 2—video 3.** Cytokinesis in the intestinal organoids.
DOI: https://doi.org/10.7554/eLife.36739.013
**Figure 2—video 4.** Furrow ingression in dissociated intestinal cells.
DOI: https://doi.org/10.7554/eLife.36739.014
**Figure 2—video 5.** Spindle assembly and cell rounding during mitosis.
DOI: https://doi.org/10.7554/eLife.36739.015
**Figure 2—video 6.** Cell rounding during mitosis in intestinal organoids.
DOI: https://doi.org/10.7554/eLife.36739.016
**Figure 2—video 7.** Chromosome movements at mitotic onset in Latrunculin A-treated organoids.
DOI: https://doi.org/10.7554/eLife.36739.017
**Figure 2—video 8.** Chromosome movements at mitotic onset in nocodazole-treated organoids.
DOI: https://doi.org/10.7554/eLife.36739.018
**Figure 2—video 9.** Chromosome movements at mitotic onset in control organoids
DOI: https://doi.org/10.7554/eLife.36739.019
**Figure 2—video 10.** Cell reinsertion behavior.
DOI: https://doi.org/10.7554/eLife.36739.020
**Figure 2—video 11.** Chromosome movements following induced mitotic exit in STLC-treated organoids.
DOI: https://doi.org/10.7554/eLife.36739.021
**Figure 2—video 12.** Chromosome movements following induced mitotic exit in STLC and Latrunculin A-treated organoids.
DOI: https://doi.org/10.7554/eLife.36739.022
**Figure 2—video 13.** Chromosome movements following induced mitotic exit in STLC and nocodazole treated organoids.
DOI: https://doi.org/10.7554/eLife.36739.023

with mitotic entry (*Figure 2C*, *Figure 2—figure supplement 1D*, *Video 3* and *Figure 2—video 5*), indicating that it was distinct from interkinetic nuclear migration, a process in which the nucleus is moved apically during interphase (interkinesis) (*Sauer, 1936*) by actin or microtubule-based forces (reviewed in [*Norden, 2017*]). Apical displacement occurred as cells adopted the rounded geometry classically associated with mitosis (reviewed in [*Théry and Bornens, 2008*]) (*Figure 2—figure supplement 1D*, *Video 1*, *Figure 2—video 5*, *Figure 2—video 6*); at metaphase and anaphase, only fine membranous processes tethered the cell to the basal surface (*Figure 2—figure supplement 1E–F*), consistent with previous observations (*Carroll et al., 2017*; *Fleming et al., 2007*; *Jinguji and Ishikawa, 1992*; *Trier, 1963*). Mitotic rounding also contributes to late stages of interkinetic nuclear migration in some systems (*Meyer et al., 2011*; *Spear and Erickson, 2012*). Therefore, we tested the importance of actin-driven mitotic rounding for apical displacement. Treatment with the actin depolymerizing small molecule Latrunculin A disrupted rounding and apical displacement (*Figure 2D,E*, *Figure 2—video 7*); in contrast, cells treated with the microtubule depolymerizing drug nocodazole rounded onto the apical surface similarly to control cells (*Figure 2D and E*,

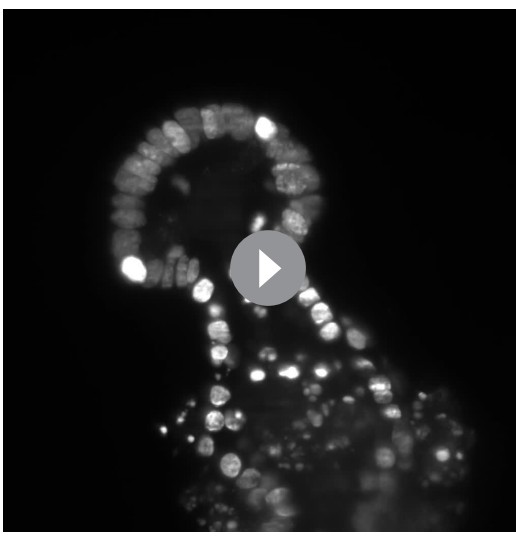

**Video 3.** Chromosome movements in intestinal organoids. H2B-mScarlet labeled organoids were imaged by SPIM using 40X objectives at 2 min time points.
DOI: https://doi.org/10.7554/eLife.36739.024

*Figure 2—video 8*, *Figure 2—video 9*). As Latrunculin-treated cells entered anaphase, the chromosome masses were positioned orthogonally to the plane of the epithelium, in contrast to the planar divisions observed in control cells (*Figure 2F*, *Figure 2—figure supplement 1G*, *Figure 2—videos 7*, *9*). This suggests that cell rounding is crucial for the normal planar orientation of the spindle in the intestine, as in some *Drosophila* epithelia (*Chanet et al., 2017*; *Nakajima et al., 2013*). Collectively, our data suggest that actin-based cell rounding displaces mitotic cells apically and is required for planar spindle orientation.

We next assessed the mechanisms that restore the basal footprint and the basal position of the nuclei after division. After division, we observed that the basal edge of nascent daughters extended a protrusive front that resembled the leading edge of migrating cells (*Figure 2G*; *Figure 2—video 10*). Therefore, we tested the contributions of the actin cytoskeleton for basal reinsertion. As actin disruption blocks the initial displacement of mitotic cells to the apical surface (*Figure 2D and E*), determining the requirements for actin in basal reinsertion required that mitotic cells be positioned on the apical surface before disrupting actin. To achieve this, we first blocked cells on the apical surface by arresting them in mitosis with the mitotic kinesin (Eg5) inhibitor S-trityl-L-cysteine (STLC). Cells arrested in mitosis did not reinsert unless mitotic exit was induced by inhibition of the spindle assembly checkpoint (SAC; Mps1 inhibitor AZ3146) or cyclin-dependent kinase (CDK; RO-3306) (*Figure 2—figure supplement 1H*, *Figure 2—video 11*). Thus, mitotic exit and reversal of CDK phosphorylation are sufficient for basal reinsertion, even in the absence of chromosome segregation.

Using this mitotic arrest and exit protocol, we tested the requirements for the actin and microtubule cytoskeletons for basal reinsertion (*Figure 2—figure supplement 1I*). When we disrupted the actin cytoskeleton and induced mitotic exit, the nucleus reformed its interphase morphology on the apical surface and the cell boundary did not protrude toward the basal surface (*Figure 2H,I*, *Figure 2—video 12*). In contrast, depolymerizing microtubules with nocodazole and inducing mitotic exit did not interfere with the ability of nuclei or the cell boundary to reach the basal surface (*Figure 2H,I*, *Figure 2—video 13*). Although actin also plays a critical role in cytokinesis, nuclei reinserted normally following inhibition of cytokinesis using the Polo-like kinase one inhibitor, BI2536 (*Lénárt et al., 2007*; *Steegmaier et al., 2007*) (*Figure 2—figure supplement 1J*), indicating that cytokinesis is dispensable for basal movement. Collectively, these data indicate that actin-driven cell elongation after mitotic exit re-establishes the interphase architecture of daughter cells.

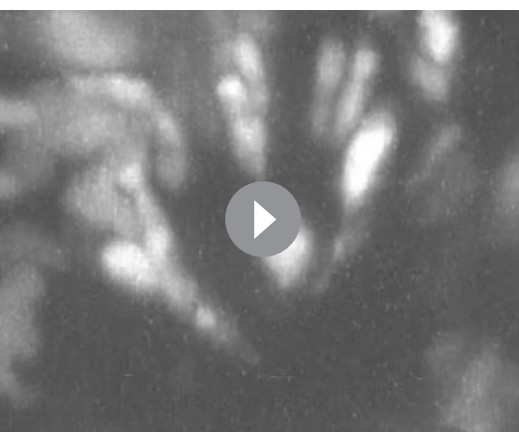

**Video 4.** Cell separation during division in the embryonic epiblast/primitive ectoderm. An E7.5 mouse embryo was imaged by SPIM with 40X objectives at 4 min time points. Cells expressing cytoplasmic RFP from a *CAGGS^{CreER}; R26^{Brainbow2.1/+}* embryo are shown.
DOI: https://doi.org/10.7554/eLife.36739.030

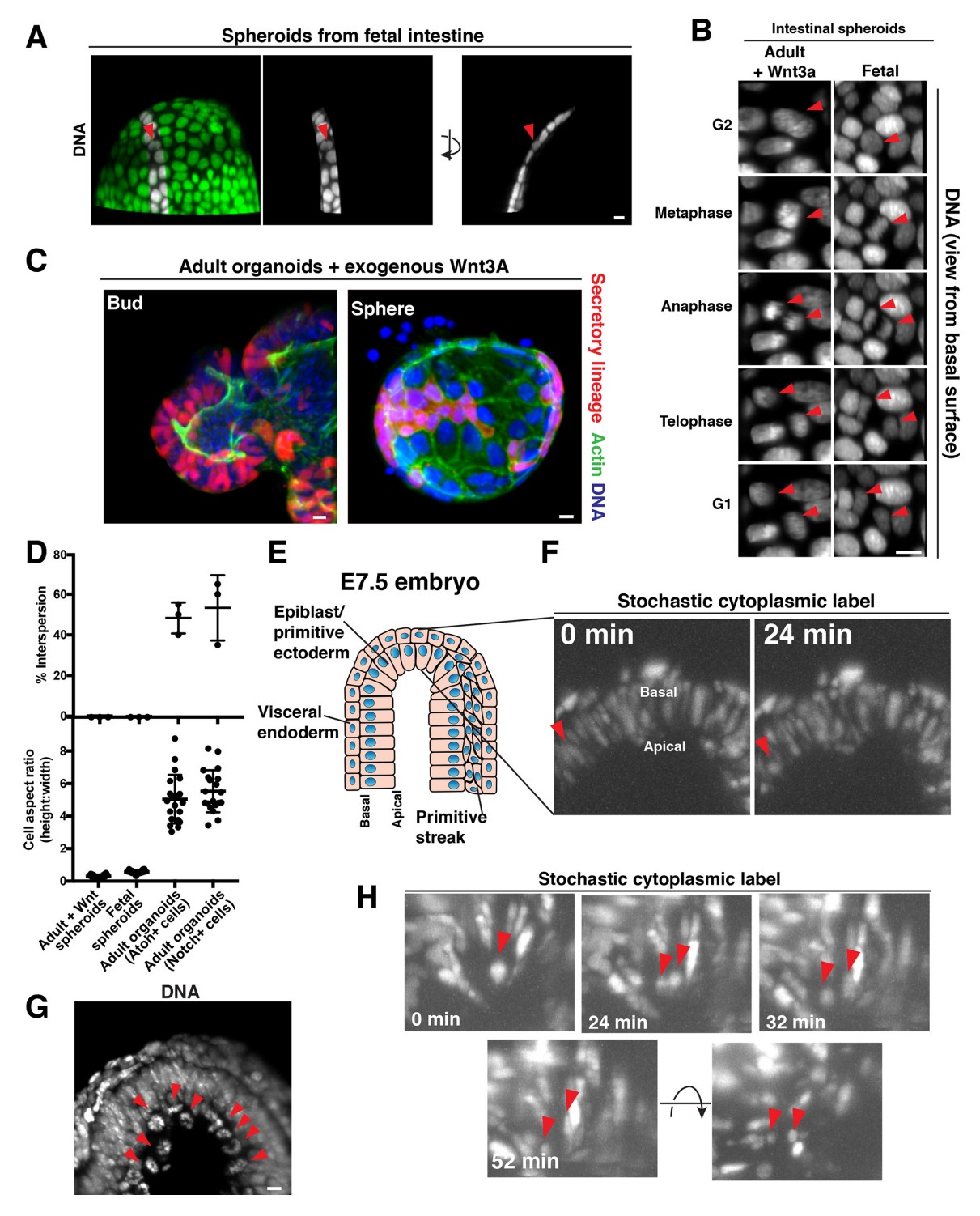

**Figure 3.** Cellular aspect ratio is a key parameter for division-coupled interspersion in the intestine and the embryo. (**A**) 3D reconstruction of a spherical organoid derived from fetal (E13.5) mouse intestine. DNA: Syto 21 dye, arrowhead: mitotic cell. (**B**) Frames from 3D reconstructed time-lapse SPIM of chromosome segregation in spheroids cultured in exogenous Wnt3a (left, DNA: H2B-GFP) or derived from fetal intestine (right, DNA: H2B-mScarlet). Arrowheads: dividing cell. Views from the basal surface are shown. (**C**) Immunofluorescence images of secretory-lineage labeled organoids (*Atoh1^{CreER}*; *R26^{RFP}*) grown with buds (left) or as spheres (right) in exogenous Wnt3a. Actin: Alexa 488-phalloidin, DNA: Hoechst 33342. Image scaled with γ adjustment. (**D**) Quantification of the frequency of division-coupled interspersion in three replicates (top, n = 20 divisions per replicate) and cellular aspect ratio (bottom, n = 20). (**E**) Cartoon depicting the embryonic portion of an E7.5 (late streak) mouse egg cylinder, distal end up. (**F**) Frames from

*Figure 3 continued on next page*

Figure 3 continued

3D reconstructed time-lapse SPIM of stochastically labeled cells (RFP + cells of *CAGGS*<sup>*CreER*</sup>; *R26*<sup>*Brainbow2.1*</sup>) in the epiblast/primitive ectoderm. Arrowhead: cell displacing to the apical surface as it enters mitosis. (**G**) 3D reconstruction of H2B-GFP embryos. Arrowheads: mitotic chromosomes. (**H**) Frames from 3D reconstructed time-lapse SPIM of stochastically labeled cells (RFP + cells of *CAGGS*<sup>*CreER*</sup>; *R26*<sup>*Brainbow2.1*</sup>) in the epiblast/primitive ectoderm. Arrowheads: dividing cell and nascent daughters, which become separated by an unlabeled cell. Scale bars, 10 μm.
DOI: https://doi.org/10.7554/eLife.36739.025
The following video and figure supplement are available for figure 3:

**Figure supplement 1.** Epithelia with low aspect ratio.
DOI: https://doi.org/10.7554/eLife.36739.026
**Figure 3—video 1.** Division in fetal spheroids with a short apical-basal axis.
DOI: https://doi.org/10.7554/eLife.36739.027
**Figure 3—video 2.** Division in fetal spheroids.
DOI: https://doi.org/10.7554/eLife.36739.028
**Figure 3—video 3.** Division in the embryonic visceral endoderm.
DOI: https://doi.org/10.7554/eLife.36739.029

## The cellular aspect ratio is a key parameter for allowing interspersion during division

Our data indicate that the displacement of cells along the elongated apical-basal axis over the course of cell division plays a role in cell interspersion. To test the importance of an elongated apical-basal axis for cell interspersion, we imaged cell behavior in spherical organoids derived from fetal intestine (*Fordham et al., 2013*; *Mustata et al., 2013*), in which cells are very short in the apical-basal dimension, and are instead elongated along the sphere circumference (*Figure 3A*, *Figure 3—video 1*). Fetal spheroids did not exhibit apical-basal mitotic movements and the daughters did not intersperse with other cells during division (*Figure 3A,B*, *Figure 3—video 1*, *Figure 3—video 2*) (50/50 divisions).

We also induced a subset of adult intestinal organoids to adopt a spherical geometry and short apical-basal axis by addition of exogenous Wnt to the medium (*Sato et al., 2011*) (*Figure 3—figure supplement 1A*). These adult spheroids also failed to exhibit apical-basal mitotic movements and the daughters did not intermix with other cells (50/50 divisions) (*Figure 3B*, *Figure 3—figure supplement 1A*). Consistent with the lack of interspersion, these spheroids contained patches of cellular progeny (*Figure 3C*), in contrast to the interspersed pattern of cell lineages observed in normal adult organoids (*Figure 1A*). As an internal control, a subset of organoids cultured in high Wnt conditions retained their budded morphology and elongated apical-basal cell shape; these organoids continued to exhibit apical displacement and the interspersed pattern of cell lineages (*Figure 3C*). This experiment, as well as our observations of adjacent progeny in the fetal spheroids, which exhibit very low expression of the Wnt reporter gene *Axin2* (*Mustata et al., 2013*), indicate that the effect of cell shape on interspersion is separable from hyperactive Wnt signaling, in contrast with previous work (*Carroll et al., 2017*). Together, these data indicate that an elongated apical-basal axis is critical for apical mitosis and cell interspersion during division.

## Apical displacement during division underlies cell interspersion in the elongated epithelium of the embryonic primitive ectoderm

Based on our data suggesting a crucial role for the cellular aspect ratio in interspersion in the organoids (*Figure 3D*), we next examined whether the mechanisms that we defined in the intestine may be relevant to other tissues with similar physical parameters. Pioneering work by *Gardner and Cockroft (1998)* revealed that cells injected into mouse blastocysts to generate chimeras become dispersed throughout the epiblast and primitive ectoderm of the post-implantation embryo. The authors proposed that this pattern might arise as a consequence of cell division, which they and others have observed occurs on the apical surface of the tissue (*Gardner and Cockroft, 1998*; *Ichikawa et al., 2013*). Therefore, we tested this prediction by performing time-lapse SPIM imaging of E7.5 (late streak-early bud) mouse embryos (*Figure 3E*), in which the epiblast/primitive ectoderm was mosaically labeled (*CAGGS*<sup>*CreER*</sup>; *R26*<sup>*Brainbow2.1*</sup>). We imaged cell divisions in these embryos for at least 3 hr and observed that divisions proceeded in a similar manner to the intestinal epithelium, with mitotic cells displacing to the apical surface as they rounded (*Figure 3F–G*). Daughter cells then

separated from one another and interspersed with unlabeled cells during cytokinesis (*Figure 3H*, *Video 4*) (8/10 divisions, n = 3 embryos from three pregnancies). Thus, daughter cells positioned on the apical surface intersperse with other cells during cytokinesis in the elongated epiblast/primitive ectoderm of the embryo, as in the adult small intestine. In contrast, the cells of the visceral endoderm (the low aspect ratio cells that surround the epiblast) did not exhibit apical displacement and daughters remain adjacent (*Figure 3—figure supplement 1B*, *Figure 3—video 3*, 12/12 divisions, n = 3 embryos from three pregnancies), consistent with classical experiments reporting outgrowth of contiguous clones in this tissue (*Gardner, 1984*; *1985*; *Lawson et al., 1991*). Thus, cell division generates distinct progeny patterns in the two layers of the early post-implantation mouse embryo, consistent with a central role for cellular aspect ratio in determining the spatial patterning of cell progeny.

## Discussion

The functions of epithelial organs rely on the concerted action of multiple cell types. As these cell types are replenished as the organ renews, they must be positioned appropriately within the tissue. In some mammalian epithelia, such as the small intestine, daughter cells derived from a common progenitor disperse throughout the tissue and intermingle with cells of other lineages, a process that plays an important role in determining local signaling environments. Previous studies have reported that intermingling of cells can occur during cell division (*Carroll et al., 2017*; *Firmino et al., 2016*; *Gardner and Cockroft, 1998*; *Higashi et al., 2016*; *Lau et al., 2015*; *Packard et al., 2013*) but the mechanism by which this occurs has not been clear. Here, we show that intermixing arises when a neighboring cell inserts between apically displaced daughter cells during cytokinesis.

The process of intermixing requires that the neighboring cell and dividing cell are positioned in such a way that the neighbor can occupy the wedge between the daughters generated by the ingressing furrow. Our data support a model in which the neighboring cell can become opportunely positioned for invasion into the cytokinetic furrow as a consequence of the cell shape changes associated with vertebrate mitosis in tissues comprised of cells with a high aspect ratio (*Figure 4*). In cells with a high aspect ratio, the actin-driven cell shape changes required for mitosis (rounding and subsequent elongation) displace the dividing cell along the apical-basal axis (*Figure 4*). As a result, an elongated interphase neighboring cell can surround the dividing cell both basally and laterally, allowing it to follow the path of the ingressing furrow between the daughters. Consistent with a key role for cell aspect ratio in interspersion behavior, reducing the aspect ratio in organoids generates patches. Live imaging of cell division in the two epithelial layers of the peri-gastrulation mouse embryo further supports a model in which cell aspect ratio is a critical parameter for determining whether cellular progeny intersperse, raising the intriguing possibility that the patterning principles that we define in the intestine may be a common feature of many mammalian epithelia.

Several lines of evidence support a model in which interspersion arises as a mechanical consequence of executing planar cell division in elongated cells, rather than being determined by developmental signaling or differential adhesion between cells. First, daughter separation is observed throughout the intestinal crypt for all progenitor cell identities: stem cells, Notch-expressing absorptive progenitors and Atoh1-expressing secretory progenitors (*Figure 1*, *Figure 1—figure supplement 1A*). However, importantly, daughter separation is a frequent but not universal event, occurring in approximately half of the divisions observed, including when observing cells of a specific lineage (*Figure 1*, *Figure 3D*). Additionally, altering cell fates, for example by inhibiting Notch signaling to cause an expansion of secretory cells, does not alter the frequency of this process (*Figure 1—figure supplement 1C,D*). In contrast, altering epithelial geometry in culture disrupts interspersion (*Figure 3*).

Since our data indicate that interspersion can arise from the execution of planar cell division coupled with the physical parameters of the tissue, it raises the possibility that the mechanisms of interspersion that we define for the intestinal epithelium may be generalizable to other vertebrate tissues with similar physical parameters. Consistent with this notion, we observed similar interspersion in the high aspect ratio epithelium of the early mouse embryo, while the surrounding low aspect ratio epithelium did not exhibit division-coupled interspersion (*Figure 3*). Several tissues across vertebrates with a high aspect ratio have also been reported to exhibit division-coupled interspersion

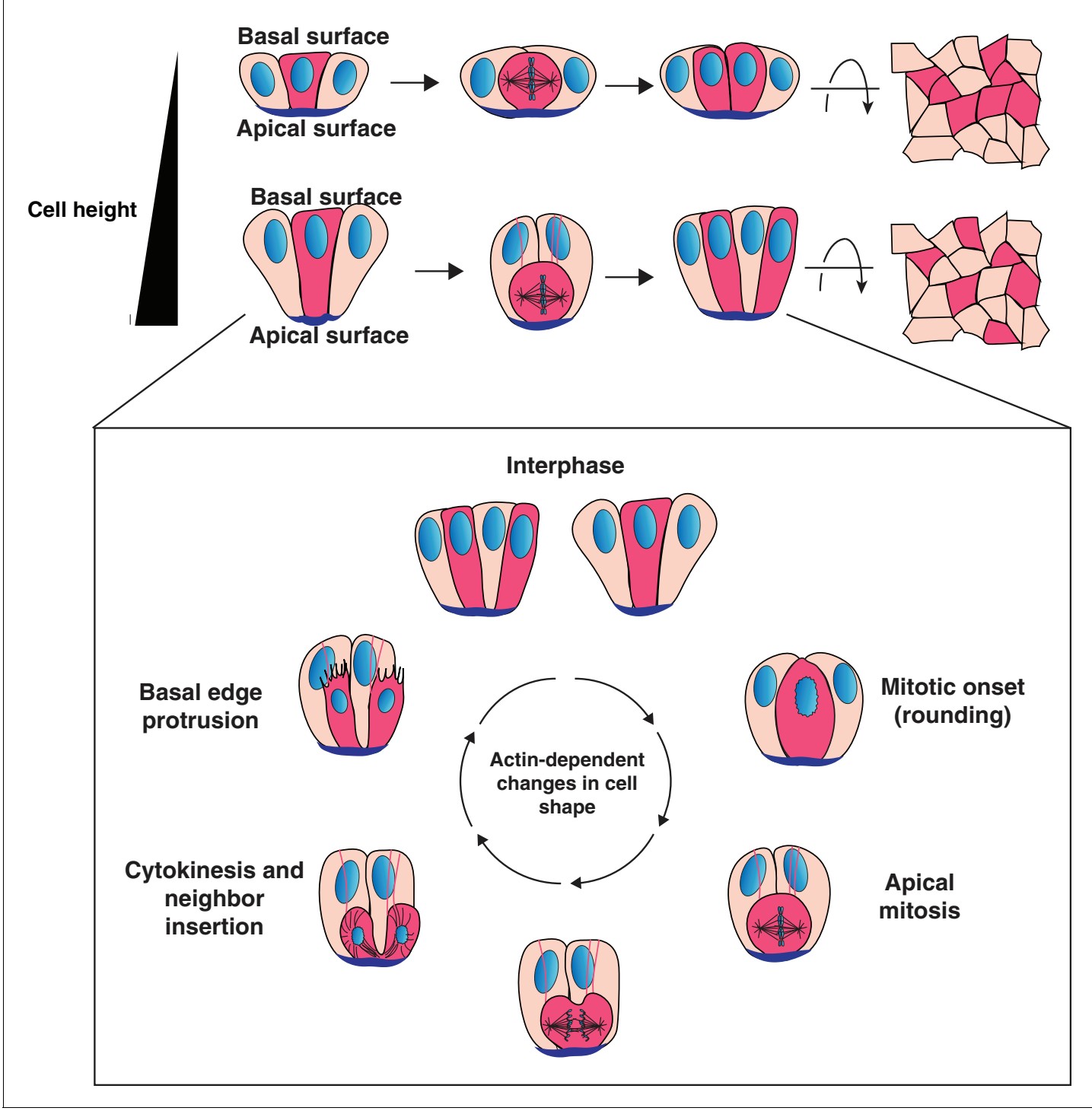

**Figure 4.** Model for cell progeny patterning in mammalian epithelia. Top: Cartoon of the influence of cell height on the relative positioning of cells derived from a given progenitor (magenta cells). Bottom: Model for interspersion of cell progeny in elongated epithelia. The basolateral surface of a dividing cell undergoes dramatic actin-dependent changes in cell shape that displace the chromosomes and cell body along the apical-basal axis. Neighboring cells can position within the ingressing cytokinetic furrow, displacing daughter cells from one another as they reinsert onto the basal surface.

DOI: https://doi.org/10.7554/eLife.36739.031

The following figure supplement is available for figure 4:

**Figure supplement 1.** Daughter cell geometries.

DOI: https://doi.org/10.7554/eLife.36739.032

(*Carroll et al., 2017*; *Firmino et al., 2016*; *Gardner and Cockroft, 1998*; *Higashi et al., 2016*; *Packard et al., 2013*). In contrast, in numerous tissues in which cells have a low aspect ratio, progeny remain adjacent and form contiguous patches, including the interfollicular epidermis (*Ouspenskaia et al., 2016*; *Rompolas et al., 2016*), MDCK cells (*Reinsch and Karsenti, 1994*), and alveolar epithelial cells (*Desai et al., 2014*). Our model raises the possibility that isolated reports of division-coupled interspersion in diverse vertebrates including frog, chick and mouse may be unified by a common physical mechanism arising from the aspect ratio of the tissue and the mechanics of cell division.

While our data indicate that cellular aspect ratio is an important parameter for interspersion, the mechanics and geometry of cytokinesis also appear to play a central role. In vertebrates, the mechanism of furrow ingression minimizes the contact between the daughters and progresses until a single apex physically connects the two cells (*Higashi et al., 2016*) (*Figure 1—figure supplement 1E*, *Figure 4—figure supplement 1*). An important component of our model is that the development of the furrow creates a position, both basally and laterally, for neighboring cells to invade and occupy. However, in contrast, during cytokinesis in Drosophila, the two daughters form a long adhesive contact between them (*Gibson et al., 2006*) (*Figure 4—figure supplement 1*), dependent on myosin II accumulation in the neighboring cells (*Herszterg et al., 2013*; *Pinheiro et al., 2017*). In this regard, it is interesting to note that Drosophila epithelia exhibit a high aspect ratio, apical mitosis and non-concentric cytokinesis, yet do not exhibit cell interspersion and form contiguous patches of progeny (*Bryant, 1970*; *Bryant and Schneiderman, 1969*; *Founounou et al., 2013*; *Gibson et al., 2006*; *Guillot and Lecuit, 2013*; *Herszterg et al., 2013*; *Meyer et al., 2011*; *Morais-de-Sá and Sunkel, 2013*). We speculate that the extended cell-cell contact formed between daughter cells in Drosophila would oppose the invasion of a neighboring cell. In the future, it will be interesting to attempt to modify the extent of interactions between daughter cells either in *Drosophila* or vertebrate epithelia and determine the effects on progeny patterning.

Broadly, since our data suggest that cell interspersion requires a set of criteria that are satisfied by many vertebrate epithelia, it is unlikely to be unique to those tissues in which it has been reported. Although our work has focused on the columnar epithelium of the small intestine, in which mitotic cell shape changes are sufficient to displace dividing cells relative to their neighbors, the numerous elongated pseudostratified epithelia that undergo apical mitosis due to interkinetic nuclear migration (reviewed in [*Norden, 2017*]) are particularly attractive candidates for division-coupled interspersion. Together, our model suggests that interspersion during cell division may be widespread across elongated vertebrate epithelia.

# Materials and methods

**Key resources table**

| Reagent type (species) or resource | Designation | Source or reference | Identifiers | Additional information |
|---|---|---|---|---|
| Strain, strain background (*Mus musculus*) | $R26^{mTmG}$ | Jackson Labs, PMID: 17868096 | MGI: 3716464 | |
| Strain, strain background (*Mus musculus*) | $Vil1^{Cre-ERT2}$ | Averil Ma lab, PMID: 15282745 | MGI: 3053826 | |
| Strain, strain background (*Mus musculus*) | $Atoh1^{CreERT}$ | Jackson labs, PMID: 16958097 | MGI: 3686985 | |
| Strain, strain background (*Mus musculus*) | $R26^{RFP}$ | Jackson Labs, PMID: 20023653 | MGI: 3809524 | |
| Strain, strain background (*Mus musculus*) | $Lgr5^{DTR-GFP}$ | de Sauvage Lab (Genentech), PMID: 21927002 | MGI: 5294798 | |

*Continued on next page*

*Continued*

| Reagent type (species) or resource | Designation | Source or reference | Identifiers | Additional information |
|---|---|---|---|---|
| Strain, strain background (*Mus musculus*) | C57BL/6J | Jackson Labs | | |
| Strain, strain background (*Mus musculus*) | $R26^{Brainbow2.1}$ | Jackson Labs, PMID: 20887898 | MGI: 164644 | |
| Strain, strain background (*Mus musculus*) | H2B-GFP | Jackson Labs, PMID:15619330 | MGI: 109836 | |
| Strain, strain background (*Mus musculus*) | $Notch1^{CreERT2(SAT)}$ | PMID: 21991352 | MGI: 5304912 | |
| Antibody | Rabbit anti-ZO-1 | Thermo Fisher | RRID:AB_2533456 | |
| Chemical compound, drug | Alexa 488-Phalloidin | Thermo Fisher | A12379 | |
| Chemical compound, drug | Hoechst 33342 | Molecular Probes | H3570 | |
| Chemical compound, drug | See Table 1 for pharmacological inhibitors | | | |
| Software, algorithm | MicroManager | Open Imaging, PMID: 20890901 | | |

## Mouse strains and lines

Adult mice of the following lines were used to generate organoids.

$R26^{mTmG/mTmG}$ (*Muzumdar et al., 2007*) (female)

$Vil1^{Cre-ERT2/+}$ (*el Marjou et al., 2004*); $R26^{mTmG/+}$ (male)

$Atoh1^{CreERT/+}$ (*Chow et al., 2006*); $R26^{RFP/+}$ (*Madisen et al., 2010*); $Lgr5^{DTR-GFP/+}$ (*Tian et al., 2011*) (female)

$Notch1^{CreERT2\ (SAT)/+}$ (*Fre et al., 2011*); $R26^{RFP/\ RFP}$ (*Madisen et al., 2010*) (female)

Fetal organoids were generated from E13.5 C57BL/6J embryos.

For imaging of cell interspersion in the intact intestine, adult $Vil1^{Cre-ERT2/+}$ (*el Marjou et al., 2004*); $R26^{Brainbow2.1/+}$ (*Snippert et al., 2010*) mice were used. Recombination was induced by oral gavage with one dose of 2.5 mg tamoxifen in corn oil 3 days before analysis.

Brainbow embryos were generated by crossing $CAGGS^{CreER/+}$ males (*Hayashi and McMahon, 2002*) to $R26^{Brainbow2.1/Brainbow2.1}$ (*Snippert et al., 2010*) females. Plugged females were injected intraperitoneally with 2.5 mg tamoxifen in corn oil at E5.5. H2B-GFP embryos were generated by crossing H2B-GFP males (*Hadjantonakis and Papaioannou, 2004*) to C57BL/6J females. Embryos were dissected at E7.5 and staged according to (*Delling et al., 2016*; *Downs and Davies, 1993*).

The strains of these mice were the same as previously described in their respective references at the time of acquisition but were subsequently maintained on mixed backgrounds after breeding between different lines. All experiments involving mice were approved by the Institutional Animal Care and Use Committee of the University of California, San Francisco (protocol #AN151723).

## Organoid preparation, dissociation and immunofluorescence

Small intestinal crypts were isolated from adult mice or E13.5 embryos and cultured in medium supplemented with human recombinant EGF, human recombinant Noggin and R-Spondin conditioned medium (ENR medium) as described (*Sato et al., 2009*). Catalog numbers for culture medium components are described in (*Mahe et al., 2013*). R-spondin and Wnt3a conditioned medium were used where indicated. Lentiviral transduction of adult organoids was performed as described (*Koo et al., 2011*). Fetal organoids were transduced according to the same protocol, but without the addition of exogenous Wnt3a to the medium at any step. For propagation, organoids were grown in 24-well plastic plates. For spinning disc imaging and immunofluorescence, organoids were grown in 96-well

**Table 1.** Small molecules used in this study.

| Molecule | Function | Source | Cat # | Final concentration |
|---|---|---|---|---|
| Nocodazole | Microtubule inhibitor | Calbiochem | 487929 | 5 µM |
| Latrunculin A | F-actin inhibitor | Calbiochem | 428026 | 4 µM |
| SiR DNA | DNA dye | Cytoskeleton Inc | CY-SC007 | 1 µM |
| Verapamil | Efflux pump inhibitor | Cytoskeleton Inc | CY-SC007 | 10 µM |
| MG132 | Proteasome inhibitor | Sigma | ML449 | 10 µM |
| STLC | Eg5 inhibitor | Sigma | 164739 | 10 µM |
| RO-3306 | CDK inhibitor | Calbiochem | 217699 | 10 µM |
| AZ3146 | Mps1 inhibitor | Tocris | 3994 | 2 µM |
| BI2536 | Plk1 inhibitor | Selleck Biochem | S1109 | 10 µM |
| Tamoxifen | Cre-ER inducer (applied for 6–16 hr in culture) | Sigma | T5648 | 1 µM ($Atoh1^{CreER}$ and $Notch1^{CreER}$) 0.1 µM ($Vil1^{CreER}$) |
| S – Blebbistatin | Myosin II inhibitor | Abcam | ab120491 | 200 µM |
| S – Blebbistatin | Myosin II inhibitor | Cayman | 13013 | 200 µM |
| Y27632 | ROCK inhibitor | Selleck | S1049 | 10 µM |
| DAPT | Gamma-secretase (Notch) inhibitor | Abcam | ab120633 | 50 µM |

DOI: https://doi.org/10.7554/eLife.36739.033

glass bottom dishes (Matriplate, Brooks). For SPIM, organoids were grown on glass coverslips which were then transferred to the SPIM imaging chamber (see below). For immunofluorescence, organoids were fixed in 4% PFA in PBS for 1 hr before blocking in 3% BSA, TBS, 0.1% Triton X-100. Primary antibody was incubated overnight at four degrees and secondary antibody was incubated for >2 hr at RT. Reagents used for immunofluorescence were as follows: rabbit anti-ZO-1 antibody (Thermo Fisher), Alexa488-Phalloidin (Thermo Fisher # A12379), Hoechst 33342 (Molecular Probes H3570).

For organoid dissociation, organoids in one well of a 24 well plate were washed once in PBS before Matrigel was manually disrupted by pipetting in TrypLE Select (Life Technologies) in the well. The plate was then incubated at 37°C for 7–8 min before additional disruption with a P200 pipette. The cell suspension was centrifuged in medium +5% fetal bovine serum at 1000 x g for 5 min. The pellet was resuspended in Matrigel, allowed to polymerize for 10 min and covered with ENR medium and immediately transferred to the microscope for imaging for 45 min – 1 hr.

## Tissue preparation for clone tracing

Animals were anesthetized by intraperitoneal (i.p.) injection of 250 mg/kg of body weight avertin (2,2,2-tribromoethanol) and transcardially perfused with 4% paraformaldehyde (PFA) in 0.1 M phosphate-buffered saline (PBS). Dissected tissues were post-fixed in 4% PFA for 3 hr at 4°C and cryoprotected in 30% sucrose in 1 × PBS overnight at 4°C. For whole mount tissue, the external smooth muscle and fat of the most proximal 3 cm of the small intestine was removed and epithelial tissue was coverslipped with ProLong Gold Antifade (P36930, Thermo Fisher Scientific). For tissue sections, tissue was embedded in OCT compound (4583, Sakura), frozen and stored at −80°C. Small intestine swiss rolls were cryosectioned at 50 µm and coverslipped with ProLong Gold Antifade. Whole mount tissue and sections were counterstained with DAPI (1:10000; D9542, Sigma) for 45 min or 15 min, respectively.

## Microscopy

For spinning disc confocal imaging, images were acquired on a Yokogawa CSU-X1 spinning disk confocal attached to an inverted Nikon TI microscope, an Andor iXon Ultra 897 EM-CCD camera, using Micro-Manager software (*Edelstein et al., 2010*). Imaging of 12 × 1 µm z-stacks was performed either at 4 min time intervals with a 40 × 1.30 NA Plan Fluor oil objective or a 20 × 0.75 NA objective, or at 20 s time intervals with a 60XA 1.20 NA Plan Apo water immersion objective.

Maximum intensity projections of 1–5 Z-stacks are shown unless otherwise noted. Point-scanning confocal imaging of intact intestines was performed using a Leica TCS SP8 X confocal microscope, with HyD and LAS X software. 0.76 μm optical sections were acquired sequentially with a 63 × 1.40 HC PL APO CS2 oil objective.

4-dimensional imaging was performed on an ASI diSPIM microscope equipped with 40 × 0.80W NA NIR-Apo water dipping objectives, Hamamatsu Flash 4.0 cameras, and 488 nm and 561 nm solid state lasers from Vortran, using a nightly build of the Micro-Manager software. The structure of the environmental control chamber is described in detail at https://valelab4.ucsf.edu/~nstuurman/proto-cols/diSPIMIncubator/. Temperature was maintained using 3 × 50 ohm resistors attached to the stainless steel incubation chamber holding the coverslip and medium, a 10 kOhm thermistor inserted in the medium and a temperature controller (TE Technology, Inc. TC-48–20). $O_2$ and $CO_2$ tensions in the medium were kept constant by flowing humidified gas underneath the sample chamber. To allow gas exchange, the sample was placed on a sandwich of 2 × 24 ×50 mm coverslip glasses in which 2 ~ 12×12 mm windows had been laser-cut and between which a piece of ~37.5 μm thick Teflon AF-2400 (a gift from BioGeneral, Inc.) was placed. Evaporation was minimized by layering mineral oil (Howard) over the sample. Organoids were imaged in ENR medium; embryos were imaged in DMEM +25% rat serum (Rockland, Inc.). 3D reconstructions were generated using a Micro-Manager plugin (https://github.com/nicost/MMClearVolumePlugin) that uses the ClearVolume library (*Royer et al., 2015*). 3D reconstructions are scaled with gamma adjustment. All imaging experiments were performed at 37°C, 5% $CO_2$, 20% $O_2$.

## Small molecules

Small molecule concentrations are described in *Table 1*. All stock solutions were prepared in DMSO. All pharmacological experiments were performed in the presence of 10 μM Verapamil to inhibit drug efflux.

## Quantification and statistical analysis

Details of statistical tests are provided in the figure legends. A statistical method of sample size calculation was not used during study design. Data were pooled from at least three biological replicates. When the observations presented were observed in less than 100% of cases, their frequency is noted in the figure, figure legend and/or text.

## Acknowledgements

We thank Meghan Morrissey, Adam Williamson, Taylor Skokan, Amnon Sharir, Tom Wald, and other members of the Vale and Klein laboratories for reagents and helpful comments on the manuscript. We thank Dyche Mullins for feedback on the manuscript, Frederic de Sauvage (Genentech) for the $Lgr5^{DTR-GFP}$ allele, and Ilia Koev (Biogene) for his gift of a piece of Teflon AF 2400. Funding for this work was provided by the Howard Hughes Medical Institute (to RDV), the Thyssen Foundation (to MD), and the Chan Zuckerberg Biohub (to LAR). Research reported in this publication was also supported by National Institute of Diabetes and Digestive and Kidney Disorders (NIDDK) and National Institute of Allergy and Infectious Diseases (NIAID) of the National Institutes of Health under grant number U01DK103147 (to ODK). KLM is a Damon Runyon Fellow supported by the Damon Runyon Cancer Research Foundation (DRG-2282–17).

## Additional information

### Funding

| Funder | Grant reference number | Author |
|---|---|---|
| Howard Hughes Medical Institute | | Kara L McKinley<br>Nico Stuurman<br>Ronald D Vale |
| National Institutes of Health | U01DK103147 | Kara L McKinley<br>David Castillo-Azofeifa<br>Ophir D Klein |

| | | |
|---|---|---|
| Chan Zuckerberg Biohub | | Loic A Royer |
| Damon Runyon Cancer Research Foundation | DRG-2282–17 | Kara L McKinley |
| Fritz Thyssen Stiftung | | Christoph Schartner<br>Markus Delling |

The funders had no role in study design, data collection and interpretation, or the decision to submit the work for publication.

## Author contributions

Kara L McKinley, Conceptualization, Formal analysis, Funding acquisition, Validation, Investigation, Visualization, Methodology, Writing—original draft, Writing—review and editing; Nico Stuurman, Formal analysis, Investigation, Methodology, Writing—review and editing; Loic A Royer, Formal analysis, Methodology; Christoph Schartner, Markus Delling, Investigation, Methodology; David Castillo-Azofeifa, Investigation, Visualization; Ophir D Klein, Resources, Supervision, Funding acquisition, Methodology, Writing—review and editing; Ronald D Vale, Conceptualization, Resources, Supervision, Funding acquisition, Methodology, Writing—review and editing

## Author ORCIDs

Kara L McKinley http://orcid.org/0000-0001-6283-9168
Nico Stuurman http://orcid.org/0000-0002-6179-8613
Christoph Schartner http://orcid.org/0000-0003-0599-3956
Ophir D Klein http://orcid.org/0000-0002-6254-7082
Ronald D Vale http://orcid.org/0000-0003-3460-2758

## Ethics

Animal experimentation: All experiments involving mice were approved by the Institutional Animal Care and Use Committee of the University of California, San Francisco (protocol #AN151723).

## Decision letter and Author response

Decision letter https://doi.org/10.7554/eLife.36739.036
Author response https://doi.org/10.7554/eLife.36739.037

## Additional files

### Supplementary files

• Transparent reporting form
DOI: https://doi.org/10.7554/eLife.36739.034

### Data availability

All data generated or analysed during this study are included in the manuscript and supporting files. Due to their large size (100s of GBs), the source movies are available upon request.

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
