## [Decision Letter]

Thank you for submitting your work entitled "Cellular aspect ratio and division mechanics govern the patterning of cell progeny in mammalian epithelia" for consideration by *eLife*. Your article has been reviewed by 3 peer reviewers, including Jody Rosenblatt as the Reviewing Editor and Reviewer #1, and the evaluation has been overseen by a Senior Editor.

Our decision has been reached after consultation between the reviewers. All the reviewers found the study interesting and compelling in its nature, given that this is an important, fundamental finding that is not well understood. However, there was general concern that the conclusions were oversold regarding the idea that tall epithelia intrinsically drive interspersion of cells. The results presented did not provide a clear mechanism for how the interdispersion takes place following cytokinesis, given that interspersion does not take place in other tall epithelia. The height of epithelia seemed correlative, using a small n for different types of epithelia or whether other epithelial cell types in the same organoid also disperse. Moreover, other in vivo components may contribute to dispersion that are lacking in the organoids; for instance the crypt curvature, which is not as well maintained in organoids, may contribute to the essential architecture required for interspersion. If so, the authors may gain more insight by filming divisions in ex vivo gut slices as well.

Because we felt that better supporting the claims stated in the title would take longer than the typical revision period of two months, the reviewers decided that it would be best turn down this manuscript. While I have summarized the main critiques of the paper, below, we include the full reviews.

*Reviewer #1:*

This beautifully written paper by McKinley and others reveals a new mechanism for how two daughter cells become interspersed through an epithelium. They show that interspersion occurs by cytokinesis at the apex of the epithelium and then re-attachment of two daughter cells basally to the matrix after they straddle a cell between them. They show that this mechanism is dependent on having a taller epithelium and that it also requires actin post-cytokinesis for cells to straddle and stretch down basally. I recommend publication of this manuscript that provides mechanistic insight to a long-observed puzzle in epithelial morphology, which is how do certain types of cells become interspersed throughout a particular epithelium. I have only a small number of questions that would be good to address:

1) The correlation between columnar epithelial height and the propensity to cause interspersion of daughter cells looks compelling but lack graphical analysis. It seems like a perfect place to include this would be in Figure 3C. Here, they could show a graphical correlation on cell height with wnt-treated organoids and the number of interspersed divisions or cells. It may be in the text but is not easy to see graphically and such a graph could make the point very clearly.

2) One thing that was not clear is whether all the cells within intestinal epithelia are interspersing or is it just the secretory lineage. This is important once we later find that the interspersion is linked to cell height. If so, we would expect all cells within the intestinal epithelium to intersperse. This would also predict that mitotic forces expected to drive cell migration up the villus will be dispersed, I expect.

3) Did the flatter visceral endoderm not intersperse in your videos? This would also lend more credence to the height as a controller.

*Reviewer #2:*

This manuscript examines how cell-types become interspersed during the development of intestinal organoids and presents evidence that this occurs as cells undergo cytokinesis at the apical surface and are then displaced from their siblings as they reintegrate towards the basal surface. This is an interesting question that takes advantage of the tractability of organoids for cell biological and live imaging approaches. While the manuscript is a useful contribution, I am not convinced that the data fully justify their conclusions. Most of the behaviours they observe, such as the apical movement and rounding of mitotic cells and planar spindle orientation have been well documented in a variety of other columnar or pseudostratified epithelia, so the main novelty lies in the mechanism of cell separation at cytokinesis. While their description is presumably accurate, it falls short of a mechanistic explanation and fails to consider the roles of the non-dividing basal neighbours or how cell junctions are remodelled during this process. As pointed out in the Discussion, divisions in the pseudostratified epithelia of the *Drosophila* imaginal discs share all of the features of the divisions in adult-derived intestinal organoids, including the polarised ingression of the cleavage furrow from basal to apical, yet the daughter cells are only very rarely separated during this process. Thus, any valid mechanistic model of cell dispersal in the intestine should be able to explain the *Drosophila* imaginal disc is different and the data as presented do not show any significant differences. The difference may lie in the behaviours of the neighbouring cells, as recent work has shown that the neighbours in the disc activate myosin contractility to pull on the apical junctions of the dividing cell during cytokinesis, which ensures that a new cell-cell junction forms between the newly-born daughters (Pinhiero et al., 2017). I therefore think that these results, useful though they are, have not really solved the question of cell dispersal and the conclusions are over-interpreted. The results are a valuable contribution to the field, and this might be publishable if the conclusions were less over-blown. For example, they could concentrate on their observations that the apical-basal height of the epithelium is a key parameter for allowing dispersal.

The authors claim that cell-cell junctions on the apical surface promote the polarized, asymmetric mitotic cell-shape changes (Subsection “Cells intersperse during cytokinesis as part of a suite of cell shape changes restricted to the basolateral surface by cell-cell contact”/Figure 2B/Figure 2—figure supplement 1A-E). For this, they measure the positioning of cortical bleb formation in mitotic cells within the tissue context and after dissociation into single cells. They show that blebbing occurs only basally in the tissue (Figure 2—figure supplement 1A) and argue that this is inhibited on the apical side by cell-cell contacts ("…cell-cell junctions on the apical surface…oppose mitotic shape changes on the apical surface"), as blebbing occurs symmetrically in dissociated cells. However, in my opinion, the proposed mechanistic link between cell-cell junction and asymmetric cell shape changes are insufficiently addressed by means of these experiments. For instance, if cell-cell junctions on the apical side have an inhibitory effect on bleb formation, why is blebbing constrained to the basal surface (Figure 2—figure supplement 1A) and does not occur on the lateral sides? Is it possible that the "fine membranous processes that tether the cell to the basal surface" (subsection “Rearrangements of the actin cytoskeleton during cell division displace dividing cells along the apical-basal axis”, first paragraph and Figure 2—figure supplement 1G-H) have never detached from the basement membrane when the cell body moved apically.

The dotted round cell outlines in Figures 2D, 2H, Figure 2—figure supplement 1L are confusing. It would be better to draw the outline of the cells based on the actual cell shape shown in the membrane signal.

Figure 2B and Figure 2—figure supplement 1E: It is not clear why the authors suddenly switch to a cell line with MT +end marker when they analyze membrane blebbing in dissociated cells? Why not stick to the cell line with a membrane marker as in previous experiments?

*Reviewer #3:*

The overarching question addressed in this study is that of how cells become interspersed within epithelia. This is a really interesting and universal open question with important implications. I was therefore really excited to get to read this paper, and unfortunately I need to admit I got a bit disappointed. The concluding statements made are bold, but the data presented, and more importantly system used for bulk of the analysis presented though quite clever, seems a little contrived. That said, this is not my field and I could easily be swayed.

The focus of this study is the mouse small intestine. Live (light-sheet and confocal etc.) imaging of small intestinal organoids in which subsets of cells are labelled in various genetic ways (inducible lineage-specific mosaic Cre lines, Confetti to mark clones etc.) is used to characterize the behavior of cells and begin to define mechanisms of cell interspersion within organoids as a proxy for the intestinal epithelium.

The authors use organoids derived from adult (tall) intestine and show that actin-, but not tubulin-, based behaviors drive cell interspersion, both cell rounding displacing cells apically at division and for subsequent re-establishment of a basal foothold. They then contrast these findings with fetal (short) intestinal-derived organoids which do not exhibit these behaviors (do they show this leads to a failure in interspersion and coherent clonal expansion within the epithelium?).

The authors nicely describe how cell interspersion arises when interphase neighboring cells wedge between apically dividing daughters during cytokinesis, and requires an elongated (tall) epithelium, and that when the cellular aspect ratio is perturbed interspersion is disrupted resulting in local clonal expansion (and the production of clonal patches).

Even though separation of daughter cells has been described previously in various tissues in vivo, it is not clear whether the mechanisms defined here in organoids are at play within the crypts/villi, and no in vivo correlations with the in vivo situation are made for any of the behaviors described for organoids. I feel this is a major weakness of the study.

---

## [Author Response]

We are writing concerning our submission to *eLife* and asking that it be considered for re-review.

We believe that this is an important paper that adds much needed cell biology and mechanism to a fundamental problem (cell dispersion). The two main findings are that 1) daughter cell separation occurs at the point of cytokinesis (which differs from other thoughts in the field), and 2) that a difference in the cell aspect ratio between interphase and mitotic cells impacts whether daughters disperse or remain together to form clonal patches. More importantly, in the intervening time since the paper was submitted, we have performed experiments that we believe squarely address the main criticisms that emerged in the decision letter, as well as extensively revising the text in response to the feedback received.

Point 1:

All the reviewers found the study interesting and compelling in its nature, given that this is an important, fundamental finding that is not well understood. However, there was general concern that the conclusions were oversold regarding the idea that tall epithelia intrinsically drive interspersion of cells. The results presented did not provide a clear mechanism for how the interdispersion takes place following cytokinesis, given that interspersion does not take place in other tall epithelia.

As discussed below in point 2c, we have extensively revised the manuscript to temper our statements regarding the pertinence of our model to other mammalian systems that we have not tested.

The comment also centers around the fact that tall *Drosophila* epithelia have not been reported to exhibit interspersion. This is an important consideration that we did not address sufficiently in our original manuscript. Addressing this point has allowed us to add depth and precision to our model for interspersion, and we are grateful for the encouragement to make these changes. Our data indicate that interspersion arises from a neighboring cell invading between the daughters during cytokinesis. This invasion depends on the process of cytokinesis generating a position between the daughter cells for the neighbor to occupy. Thus, our model suggests that the creation of a minimal contact between the daughters is central to the interspersion behavior; extensive contact/adhesions between the daughters during cytokinesis would oppose the invasion of the neighbor. Importantly, extensive literature has demonstrated that *Drosophila* cytokinesis generates a long adhesive contact between the daughters (Gibson, Patel, Nagpal, and Perrimon, 2006; Herszterg, Leibfried, Bosveld, Martin, and Bellaiche, 2013; Pinheiro et al., 2017). By our observations and model, one would expect this geometry to oppose neighbor cell invasion. In contrast, a minimal vertex connection is created between daughters during cytokinesis in our system and broadly in vertebrates (analyzed carefully in (Higashi, Arnold, Stephenson, Dinshaw, and Miller, 2016), please also see our new figures – Figure 4—figure supplement 1 and Figure 1—figure supplement 1E). Thus, the neighboring cell can wedge between the daughters.

We apologize for our superficial discussion of this important consideration in the previous draft, and we appreciate the opportunity to include these important refinements of our model, now included in several points in the text, and addressed carefully in the second to last paragraph in the Discussion. Naturally, the differences between cell interspersion in *Drosophila* and vertebrates begs for further experiments to test our hypothesis. This would be best accomplished by somehow modifying the extent of contact between daughters during cytokinesis. We describe this possibility in the paper as a possible future direction in the Discussion. However, this is clearly beyond the scope of our current study, both because of the complexity of the experiment and time to pioneer a completely new study of comparing and manipulating vertebrate and invertebrate systems.

Point 2:

The height of epithelia seemed correlative, using a small n for different types of epithelia or whether other epithelial cell types in the same organoid also disperse.

We have now performed the additional experiments requested by the reviewers to further support our model for the role of cell height, as well as rewriting the text to focus on the specific epithelia we have analyzed. Our paper now analyzes both epithelia of the embryo, all cell lineages in the organoid, as well as several alternative organoid systems (adult versus fetal, low and high Wnt conditions). The new additions are as follows:

a) Reviewers 1 and 3 suggested that we examine the visceral endoderm of the embryo. We have examined cell divisions in the visceral endoderm of the embryo using embryos in which the DNA is labeled with H2B-GFP. In contrast to the elongated epiblast/primitive ectoderm, the endodermal cells do not intersperse during division in this low aspect ratio tissue. The new figure 3—figure supplement 1B, and is accompanied by a new video (Figure 3—video 3).

b) We have now demonstrated that all cell lineages in the intestinal crypt exhibit interspersion, as expected based on our model indicating that aspect ratio, rather than e.g. cell identity, is a key parameter for this behavior. Previously, we only analyzed interspersion in the stem cell lineage (Lgr5-expressing) and the secretory (Atoh1-expressing) lineage. We have now analyzed Notch1-expressing cells, which represents all non-secretory cell populations, and found that these cells also intersperse at a similar frequency to secretory cells. Together, these reporters cover all cell lineages in the crypt epithelium. (New Figure 1—figure supplement 1A).

c) We have been careful to rewrite the text to state conclusions based on the subset of mammalian epithelia that we have directly tested. We have tempered our conclusions with regard to breadth in the title, abstract and throughout the text to make it clear that the principles that we have uncovered may pertain to several (not necessarily all) mammalian epithelia.

Point 3:

Moreover, other in vivo components may contribute to dispersion that are lacking in the organoids; for instance the crypt curvature, which is not as well maintained in organoids, may contribute to the essential architecture required for interspersion. If so, the authors may gain more insight by filming divisions in ex vivo gut slices as well.

We now demonstrate in vivo relevance of interspersion in the intestine. Following reviewer 3’s suggestion, we analyzed fixed intestines labeled for three days with the stochastic multicolor reporter (often referred to as confetti/Brainbow) to determine if interspersion of progeny is occurring in vivo. We found that cells derived from a given progenitor also intersperse with other cells in vivo (new Figure 1E). Live cell imaging of explant cultures of the intact intestine is not robust for our long-term imaging experiments at this point in time. In our own experience using the most recent protocols from the literature, intestinal explant cultures do not remain healthy for more than a few hours, and we do not wish to report results from less-than-healthy tissue. Therefore, we chose to do the experiment in the setting of the intact animal, where the continued health of the tissue is ensured. We appreciate the opportunity to include these data, as we think it is powerful to demonstrate that interspersion is relevant in vivo, and also that interspersion can be isolated from the chemical, mechanical and electrical cues present in vivo by using the minimal organoid system.

Reviewer #1:

[…] 1) The correlation between columnar epithelial height and the propensity to cause interspersion of daughter cells looks compelling but lack graphical analysis. It seems like a perfect place to include this would be in Figure 3C. Here, they could show a graphical correlation on cell height with wnt-treated organoids and the number of interspersed divisions or cells. It may be in the text but is not easy to see graphically and such a graph could make the point very clearly.

Thank you for this suggestion. We have now added these data as Figure 3D.

2) One thing that was not clear is whether all the cells within intestinal epithelia are interspersing or is it just the secretory lineage. This is important once we later find that the interspersion is linked to cell height. If so, we would expect all cells within the intestinal epithelium to intersperse. This would also predict that mitotic forces expected to drive cell migration up the villus will be dispersed, I expect.

Thank you for this suggestion. We have now included these data as Figure 1—figure supplement 1A. Please see point 2b in the response to the editorial summary, above.

3) Did the flatter visceral endoderm not intersperse in your videos? This would also lend more credence to the height as a controller.

Thank you for this suggestion. We have now included these data as Figure 3—figure supplement 1B. Please see point 2a in the response to the editorial summary, above.

Reviewer #2:

This manuscript examines how cell-types become interspersed during the development of intestinal organoids and presents evidence that this occurs as cells undergo cytokinesis at the apical surface and are then displaced from their siblings as they reintegrate towards the basal surface. This is an interesting question that takes advantage of the tractability of organoids for cell biological and live imaging approaches. While the manuscript is a useful contribution, I am not convinced that the data fully justify their conclusions. Most of the behaviours they observe, such as the apical movement and rounding of mitotic cells and planar spindle orientation have been well documented in a variety of other columnar or pseudostratified epithelia, so the main novelty lies in the mechanism of cell separation at cytokinesis. While their description is presumably accurate, it falls short of a mechanistic explanation and fails to consider the roles of the non-dividing basal neighbours or how cell junctions are remodelled during this process. As pointed out in the Discussion, divisions in the pseudostratified epithelia of the Drosophila imaginal discs share all of the features of the divisions in adult-derived intestinal organoids, including the polarised ingression of the cleavage furrow from basal to apical, yet the daughter cells are only very rarely separated during this process. Thus, any valid mechanistic model of cell dispersal in the intestine should be able to explain the Drosophila imaginal disc is different and the data as presented do not show any significant differences. The difference may lie in the behaviours of the neighbouring cells, as recent work has shown that the neighbours in the disc activate myosin contractility to pull on the apical junctions of the dividing cell during cytokinesis, which ensures that a new cell-cell junction forms between the newly-born daughters (Pinhiero et al., 2017).

We have now significantly rewritten the text and added new figures to address these important points (see also our response to point 1 in the editorial summary). Thanks to the reviewer’s comment, we realize that we over-emphasized cell aspect ratio and underplayed the central role that the geometry of cytokinesis plays in this process. In particular, a cornerstone of our paper is that a neighboring cell invades basally and laterally between the nascent daughters during cytokinesis. This can occur because the process of cytokinesis creates a minimal contact between the daughters, which allows an elongated neighboring cell to position between the daughters and displace them from one another. Thus, significant contact between daughter cells would be expected to prevent such intercalation. As the reviewer points out, extensive work in the literature has demonstrated that, during *Drosophila* cytokinesis, a long new adhesive contact forms between nascent daughters. This contrasts with the vertex contact between daughters in our system, and broadly observed during cytokinesis in vertebrate systems. The zippering of daughter cells together in the *Drosophila* system would be expected to represent a significant obstacle to invasion of a neighboring cell.

We realize, like any study, that this difference opens up questions for further experimentation. Probably the most relevant experiments would be to modify the degree of interaction between the daughter cells (decreasing it in *Drosophila* or increasing in the intestinal epithelium) as this would allow predictions of this model to be tested. However, we believe that strategies for manipulating cytokinesis in this manner are not entirely evident and overall this work is beyond the scope of this particular paper. However, we do now raise this possibility in the Discussion of this article.

We are grateful for the reviewer’s comment, which encouraged us to discuss the nature of the invasion mechanism in greater detail, helping us to hone in on the importance of cytokinesis geometry and minimal daughter-daughter contact in allowing interspersion to occur. We recognize that our discussion of the comparison with *Drosophila* in the previous draft was superficial and deserved more careful consideration, which we have now tried to do. We appreciate the opportunity to expand this discussion, which presented an opportunity to add more depth to our model. Indeed, the reviewer’s comment and the comparison with *Drosophila* has led us to re-craft the text and present our model more precisely.

I therefore think that these results, useful though they are, have not really solved the question of cell dispersal and the conclusions are over-interpreted. The results are a valuable contribution to the field, and this might be publishable if the conclusions were less over-blown. For example, they could concentrate on their observations that the apical-basal height of the epithelium is a key parameter for allowing dispersal.

Overall, in addition to addressing the important considerations of the *Drosophila* system as described above, we have now extensively revised the text to temper our conclusions and to focus throughout the paper on the mammalian systems for which we have direct data. As the reviewer suggests, we have focused the revised manuscript on our observations that the height of the epithelium is a key parameter for allowing dispersal, as well as incorporating discussion of the geometry of cytokinesis as described above.

The authors claim that cell-cell junctions on the apical surface promote the polarized, asymmetric mitotic cell-shape changes (Subsection “Cells intersperse during cytokinesis as part of a suite of cell shape changes restricted to the basolateral surface by cell-cell contact”/Figure 2B/Figure 2—figure supplement 1A-E). For this, they measure the positioning of cortical bleb formation in mitotic cells within the tissue context and after dissociation into single cells. They show that blebbing occurs only basally in the tissue (Figure 2—figure supplement 1A) and argue that this is inhibited on the apical side by cell-cell contacts ("…cell-cell junctions on the apical surface…oppose mitotic shape changes on the apical surface"), as blebbing occurs symmetrically in dissociated cells. However, in my opinion, the proposed mechanistic link between cell-cell junction and asymmetric cell shape changes are insufficiently addressed by means of these experiments. For instance, if cell-cell junctions on the apical side have an inhibitory effect on bleb formation, why is blebbing constrained to the basal surface (Figure 2—figure supplement 1A) and does not occur on the lateral sides? Is it possible that the "fine membranous processes that tether the cell to the basal surface" (subsection “Rearrangements of the actin cytoskeleton during cell division displace dividing cells along the apical-basal axis”, first paragraph and Figure 2—figure supplement 1G-H) have never detached from the basement membrane when the cell body moved apically.

We appreciate these points. As the blebbing is not a central component of the paper, we have now removed these data to avoid any confusion.

The dotted round cell outlines in Figures 2D, 2H, Figure 2—figure supplement 1L are confusing. It would be better to draw the outline of the cells based on the actual cell shape shown in the membrane signal.

Thank you for bringing this to our attention. We have now replaced these outlines with arrowheads, and show the membrane signal separately.

Figure 2B and Figure 2—figure supplement 1E: It is not clear why the authors suddenly switch to a cell line with MT +end marker when they analyze membrane blebbing in dissociated cells? Why not stick to the cell line with a membrane marker as in previous experiments?

We apologize for not including this information. EB3-GFP organoids were used to facilitate identification of organoids in mitosis after dissociation, when they cannot be identified by membrane morphology as all dissociated cells are spherical. We have now clarified this important point in the figure legends for Figure 2B (Figure 2—figure supplement 1E was removed, as described in the previous point). We thank the reviewer for catching this omission.

Reviewer #3:

The overarching question addressed in this study is that of how cells become interspersed within epithelia. This is a really interesting and universal open question with important implications. I was therefore really excited to get to read this paper, and unfortunately I need to admit I got a bit disappointed. The concluding statements made are bold, but the data presented, and more importantly system used for bulk of the analysis presented though quite clever, seems a little contrived. That said, this is not my field and I could easily be swayed.The focus of this study is the mouse small intestine. Live (light-sheet and confocal etc.) imaging of small intestinal organoids in which subsets of cells are labelled in various genetic ways (inducible lineage-specific mosaic Cre lines, Confetti to mark clones etc.) is used to characterize the behavior of cells and begin to define mechanisms of cell interspersion within organoids as a proxy for the intestinal epithelium.The authors use organoids derived from adult (tall) intestine and show that actin-, but not tubulin-, based behaviors drive cell interspersion, both cell rounding displacing cells apically at division and for subsequent re-establishment of a basal foothold. They then contrast these findings with fetal (short) intestinal-derived organoids which do not exhibit these behaviors (do they show this leads to a failure in interspersion and coherent clonal expansion within the epithelium?).

We show that organoids with short cells (fetal and grown in high Wnt) lead to a failure in interspersion and coherent clonal expansion. These data are included in Figure 3B and C and new Figure 3D.

The authors nicely describe how cell interspersion arises when interphase neighboring cells wedge between apically dividing daughters during cytokinesis, and requires an elongated (tall) epithelium, and that when the cellular aspect ratio is perturbed interspersion is disrupted resulting in local clonal expansion (and the production of clonal patches).Even though separation of daughter cells has been described previously in various tissues in vivo, it is not clear whether the mechanisms defined here in organoids are at play within the crypts/villi, and no in vivo correlations with the in vivo situation are made for any of the behaviors described for organoids. I feel this is a major weakness of the study.

We appreciate this important point. As described in the editorial summary, point 3, above, we have now shown that interspersion of progeny also occurs in vivo. This is an important addition to the paper, and we thank the reviewer for bringing it to our attention.